# Effect of Smoking on Treatment Efficacy and Toxicity in Patients with Cancer: A Systematic Review and Meta-Analysis

**DOI:** 10.3390/cancers14174117

**Published:** 2022-08-25

**Authors:** Marie Bergman, Georgios Fountoukidis, Daniel Smith, Johan Ahlgren, Mats Lambe, Antonios Valachis

**Affiliations:** 1Department of Oncology, Hospital of Karlstad, 652 30 Karlstad, Sweden; 2Department of Oncology, Faculty of Medicine and Health, Örebro University, 702 81 Örebro, Sweden; 3Clinical Epidemiology and Biostatistics, School of Medical Sciences, Örebro University, 702 81 Örebro, Sweden; 4Regional Cancer Centre, Mid-Sweden Health Care Region, 751 22 Uppsala, Sweden; 5Department of Medical Epidemiology and Biostatistics, Karolinska Institute, 104 35 Stockholm, Sweden

**Keywords:** smoking, cancer, efficacy, toxicity, meta-analysis

## Abstract

**Simple Summary:**

The impact of smoking on cancer treatment efficacy and toxicity regardless of cancer type was investigated in this meta-analysis. Smoking during radiotherapy/chemoradiotherapy was associated with worse outcomes and a higher risk for toxicity. Smoking during treatment with EGFR tyrosine kinase inhibitors in lung cancer patients was associated with a worse prognosis, whereas smoking was associated with better outcomes in patients treated with checkpoint inhibitors. No association between smoking and treatment efficacy of chemotherapy was observed, though with low certainty of evidence. Our results can be used by oncology and radiotherapy staff to give patients more convincing information on the benefits that can be derived from smoking cessation before cancer treatment.

**Abstract:**

Aim: The aim of the present systematic review and meta-analysis was to summarize the current evidence on the potential impact of smoking during cancer treatment on treatment efficacy and toxicity irrespective of cancer type. Methods: A systematic literature search was performed using two electronic databases for potentially eligible studies. Only studies based on multivariable analysis for the association between smoking, compared to non-smokers (never or former), and treatment efficacy or toxicity were included. Pooled Hazard Ratios (HRs) or Odds Ratios (ORs) and corresponding 95% Confidence Intervals (CIs) were estimated through random-effects meta-analyses. Results: In total, 97 eligible studies were identified, of which 79 were eligible for the pooled analyses. Smoking during radiation therapy, with or without chemotherapy, was associated with an increased risk of locoregional recurrence (pooled HR: 1.56; 95% CI: 1.28–1.91 for radiation therapy; pooled HR: 4.28; 95% CI: 2.06–8.90 for chemoradiotherapy) and worse disease-free survival (pooled HR: 1.88; 95% CI: 1.21–2.90 for radiation therapy; pooled HR: 1.92; 95% CI: 1.41–2.62 for chemoradiotherapy) as well as a higher risk for radiation-induced toxicity (pooled OR: 1.84; 95% CI: 1.32–2.56 for radiation therapy; pooled OR: 2.43; 95% CI: 1.43–4.07 for chemoradiotherapy) with low-to-moderate certainty of evidence. Smoking during treatment with EGFR tyrosine kinase inhibitors (EGFR-TKIs) in patients with lung cancer was associated with worse progression-free survival compared to non-smokers (pooled HR: 1.43; 95% CI: 1.14–1.80; moderate certainty of evidence), whereas smoking was associated with improved progression-free survival in patients treated with checkpoint inhibitors (HR: 0.70; 95% CI: 0.58–0.84; moderate certainty of evidence). No statistically significant associations were observed between smoking and treatment efficacy or toxicity to chemotherapy. Conclusion: The present meta-analysis confirms earlier evidence of the negative impact of smoking during radiation therapy, with or without chemotherapy, on treatment efficacy and radiation-induced toxicity as well as a negative impact of smoking on the efficacy of EGFR-TKIs and a positive impact on the efficacy of checkpoint inhibitors. The evidence is too weak to draw firm conclusions on the potential association between smoking and chemotherapy, whereas there is no evidence for pooled analyses regarding other types of systemic oncological therapy.

## 1. Introduction

Smoking (referred to tobacco use through cigarette smoking) represents a major risk factor for developing cancer with an estimated one of five cancer cases being directly caused by smoking [1]. The association between smoking and cancer is strongest for lung and laryngeal cancer, followed by other head and neck malignancies and cancer of the upper digestive tract [1,2].

Continued smoking after cancer diagnosis has also been found to be a strong predictor of cancer-specific and overall mortality in several types of cancer [3,4,5,6]. Current evidence suggests several potential explanations for how continued smoking can compromise survival including an increased risk for second primary cancer [7,8,9], increased risk for postoperative complications [10,11], higher risk for non-cancer-related deaths [12,13], and a detrimental effect on both radiotherapy [14,15] and systemic therapy [16,17,18,19].

Results from several site-specific cancer meta-analyses indicate that smoking during radiotherapy has a negative impact on treatment efficacy and is associated with increased toxicity [14,15]. However, pooled evidence on the impact of smoking on radiotherapy as treatment modality per se irrespective of the type of cancer is lacking.

The evidence on the potential impact of smoking on systemic therapy is less pronounced than that on radiotherapy and scarce [16,17,18,19]. Furthermore, the current evidence has mostly been summarized in terms of cancer-specific or overall survival rather than on outcomes that could reflect a treatment-specific efficacy as objective response rates for systemic therapies, local recurrences for radiotherapy, or a direct detrimental effect on toxicity.

The aim of the present review and meta-analysis was to summarize current evidence on the effect of smoking on the efficacy of oncological treatment, including radiotherapy and systemic therapy, and toxicity in patients with cancer.

## 2. Materials and Methods

This systematic review and meta-analysis was performed in accordance with PRISMA guidelines. The study protocol was prospectively registered in the PROSPERO database (CRD42020175724).

### 2.1. Search Strategy

We performed an electronic search using the following keywords in different searching algorithms: tobacco use, smoking, cancer, chemotherapy, radiotherapy, molecular targeted therapy, immunotherapy, endocrine therapy.

### 2.2. Information Sources

Three electronic databases (PubMed, ISI Web of Science, EMBASE) were searched using the following limitations: from January 1990 to May 2022 and published in the English language. In addition, a manual search based on the reference lists of the eligible studies and relevant systematic reviews and meta-analyses on the topic was performed to find additional studies.

### 2.3. Eligibility Criteria

Studies were considered eligible if they fulfilled all the following criteria:(1)Included cancer patients regardless of tumor site or type;(2)Presented data on smoking during a specific oncological therapy irrespective of the type of therapy (external radiotherapy or brachytherapy, chemotherapy, targeted therapies, immunotherapy, endocrine therapy);(3)Presented data on the association between smoking and treatment efficacy with one of the following outcomes: pathologic complete response (pCR), objective response rate (ORR), locoregional recurrence (LRR; for radiotherapy), event-free survival (EFS; for neoadjuvant setting), disease-free survival (DFS; for adjuvant setting), progression-free survival (PFS; for metastatic setting), treatment-related toxicity;(4)Compared the treatment efficacy in patients who smoked vs. non-smokers (never or former).

Only studies that used multivariable analyses to examine the association between smoking and treatment efficacy or toxicity were included to minimize the risk for confounding bias. Similarly, we included only randomized trials where smoking status was used as a stratifying factor for randomization or where a multivariable analysis was used to investigate the association between smoking and treatment efficacy or toxicity.

We excluded meta-analyses, studies without comparison between smokers vs. non-smokers or with inadequate description of smoking, studies analyzing only cancer-specific or overall survival as outcomes, and studies without separate analyses for the effect of smoking in specific treatments.

The PICO approach for the systematic review was as follows:P: Patients with cancer treated with chemotherapy, radiotherapy, targeted therapies, immunotherapy, or endocrine therapy;I: Smoking at the time of therapy initiation and during therapy;C: Non-smokers or prior smokers (non-smokers at the time of therapy initiation);O: Treatment efficacy in terms of time-to-event measures (LRR for radiotherapy or chemoradiotherapy, EFS for neoadjuvant setting, DFS for adjuvant treatment, PFS for treatment in the metastatic setting), pCR, ORR, and toxicity due to treatment.

### 2.4. Selection Process

Two researchers (M.B., G.F.) independently identified potentially eligible studies through the searching strategies, and consensus on the eligible studies was reached in discussion with a third researcher (A.V.).

### 2.5. Data Collection Process

Two researchers (M.B., G.F.) independently collected relevant data from each eligible study using a prespecified form. The following data were collected: 1st author’s name, publication journal, year of publication, study origin, type of study, enrollment period, number of included patients, follow-up time, cancer type, definition of smoking, treatment strategy, outcome of interest based on smoking (Odds Ratios (ORs) and corresponding 95% Confidence Intervals (CIs) for categorical outcomes; Hazard Ratios (HRs) and corresponding 95% CI for time-to-event outcomes).

### 2.6. Outcomes

The following outcomes were considered as relevant for the association between smoking and radiotherapy or chemoradiotherapy: LRR, DFS, and radiation-induced toxicity.

The following outcomes were considered as relevant for the association between smoking and systemic therapy: pCR, ORR, DFS, PFS, and treatment-related toxicity.

All treatment-related toxicities of at least grade 2 according to internationally accepted scales such as CTCAE (any version) or RTOG were considered as relevant.

For all efficacy outcomes, the definition that was used in each study was adopted.

### 2.7. Data Synthesis

We computed the pooled Hazard Ratio (HR; for outcomes: LRR, DFS, and PFS) and Odds Ratio (OR; for toxicity) using random-effects meta-analyses. The choice of random-effects models was based on the nature of our statistical inference rather than on the assessment of statistical heterogeneity. Furthermore, the statistical assessment of heterogeneity in each pooled analysis was calculated using the Cochran’s Q test and is presented in each forest plot as Q and *p*-value. The HR and OR from primary studies represented the effects of current smoking compared with previously or never smoking, though the reference level varied across studies. We therefore inverted several effect sizes by taking their reciprocal, so that previous/never smoking represented the reference level across all studies. Pooled effects were computed for prespecified combinations of outcomes, treatment, and cancer type including estimates from at least two studies. We used the logarithm of the HR and OR, which are both known to have normally distributed sampling distributions. Corresponding standard errors were required to weight the effect sizes using the inverse variance method. If these were not provided directly, they were estimated from reported confidence intervals (see Appendix A for details).

Empirical Bayes estimation was used to compute pooled summary effects, which were then exponentiated for presentation on the original scale. Where pooled estimates utilized multiple effect sizes from the same study, a sandwich-type estimator was used to construct a cluster-robust variance–covariance matrix, which included a small sample adjustment [20]. The risk for publication bias for pooled analyses with an adequate number of studies was assessed through funnel plots. All statistical analyses were performed in R (Version 4.1.0) using R studio (Version 1.4.1717), relying heavily on the package ‘metafor’ [21].

### 2.8. Grading the Certainty of Evidence

The GRADE approach was used to rate quality of cumulative evidence for each outcome of this systematic review [22]. The GRADE approach specifies four levels of quality that vary from very low to high.

## 3. Results

### 3.1. Literature Search

The searching of electronic databases PubMed and ISI Web of Knowledge identified a total of 11,360 studies. After reviewing the title and abstract, 385 potentially eligible studies were selected. An additional 32 studies were selected as potentially eligible through manual searching based on the reference lists of the eligible studies and relevant systematic reviews and meta-analyses.

The full text was retrieved for potentially eligible studies, and inclusion and exclusion criteria were applied to them. In total, 97 studies were considered eligible to be included in the review (Figure 1).

For each pooled analysis, all eligible studies that presented adequate data regarding type of treatment (radiation therapy, chemoradiotherapy, chemotherapy, targeted therapy, or immunotherapy) and efficacy measures (LRR, DFS, PFS, or treatment-related toxicity) were included.

A pooled analysis was performed if there were at least three studies with adequate data for each combination of treatment type and effect measure. Based on this statistical requirement, some treatment types with few eligible studies (endocrine therapy, Bacillus Calmette–Guérin (BCG) instillations) as well as some efficacy measures of potential interest (pCR, ORR) were excluded.

When applying the above-mentioned restrictions regarding pooled analyses, 78 studies were eligible for the pooled analyses.

### 3.2. Characteristics of Eligible Studies

Of 97 eligible studies, 81 (84%) were retrospective, 13 (13%) were prospective and 3 (3%) were randomized studies. A total of 36 (37%) studies included patients with non-small-cell lung cancer (NSCLC) followed by 18 (19%) studies with head and neck tumors and 12 (12%) with breast cancer. Five (5%) studies included patients with various types of cancer.

The number of patients included in the studies varied between 48 and 8649 patients. A total of 53 (55%) studies presented data on PFS, 29 (30%) on treatment-related toxicity, 15 (15%) on LRR, 12 (12%) on DFS, 6 (9%) on ORR and 4 (6%) on pCR.

A summary description of the characteristics of eligible studies is presented in Appendix A.

### 3.3. Smoking during Radiation Therapy

The impact of smoking during radiation therapy on treatment efficacy could be analyzed for two effect measures: LRR and DFS.

Ten studies (13,276 patients) presented data on smoking during radiation therapy and the risk of LRR. The pooled HR was 1.56 (95% CI: 1.28–1.91; Figure 2a), showing an increased risk of LRR in smokers compared to non-smokers during radiation therapy.

Nine studies (11,185 patients) presented data on DFS after radiation therapy in smokers compared to non-smokers. Smoking during radiation therapy led to worse DFS with a pooled HR of 1.88 (95% CI: 1.21–2.90; Figure 2b).

### 3.4. Smoking during Chemoradiotherapy

Three studies (339 patients; 68 with anal cancer, 271 with head and neck cancer) presented data on the risk of LRR in smokers during chemoradiotherapy compared to non-smokers. Smoking was associated with an increased risk of LRR (pooled HR: 4.28; 95% CI: 2.06–8.90; Figure 3a).

Seven studies (2096 patients) presented data on smoking during chemoradiotherapy in relation to DFS. The pooled analysis showed worse disease-free survival for smokers (pooled HR: 1.92; 95% CI: 1.41–2.62; Figure 3b).

### 3.5. Smoking during Chemotherapy

Six studies with seven comparisons (one study presented separate analyses for two different chemotherapy indications) including a total of 1489 patients reported data on PFS for smokers during chemotherapy compared to non-smokers. Smoking during chemotherapy was not associated with PFS (pooled HR: 1.22; 95% CI: 0.63–2.36; Appendix A).

### 3.6. Smoking during Targeted Therapies

Our literature search could identify only one category of targeted therapy (epidermal growth factor receptor tyrosine kinase inhibitors; EGFR TKIs) where there were studies with adequate data investigating the potential impact of smoking on treatment effect.

Sixteen studies (7682 patients) were included in a pooled analysis of smoking and PFS in patients treated with EGFR TKIs. Smoking during treatment with EGFR TKIs was associated with worse PFS with a pooled HR of 1.46 (95% CI: 1.21–1.77; Appendix A).

### 3.7. Smoking during Immunotherapy with Checkpoint Inhibitors

Eleven studies including a total of 4568 patients presented data on smoking during immunotherapy (PD-1 or PD-L1 inhibitors) and treatment effect. A statistically significant difference in favor of smokers was observed (pooled HR: 0.70; 95% CI: 0.61–0.82; Appendix A). All but one study included in this pooled analysis were retrospective. When the only randomized study was excluded from the pooled analysis, the pooled HR remained statistically significant (pooled HR: 0.70; 95% CI: 0.58–0.84).

### 3.8. Smoking and Treatment-Related Toxicity

The impact of smoking on treatment-related toxicity could be analyzed for three different types of cancer treatment: radiation therapy, chemotherapy, and chemoradiotherapy.

Fifteen studies (6776 patients) included data on radiation-induced toxicity (regardless of toxicity) in smokers during radiation therapy compared to non-smokers. Smoking during radiation therapy led to an increased risk of radiation-induced toxicity with an OR of 1.84 (95% CI: 1.32–2.56; Figure 4a).

Nine studies (3307 patients) presenting data from 13 different toxicities (one study presented separate data on three side effects and two on two side effects) were included in a pooled analysis of how smoking affects the risk of chemotherapy-induced toxicity. No statistically significant difference in toxicity risk between smokers and non-smokers could be found (pooled OR: 0.92; 95% CI: 0.53–1.60; Figure 4b).

Four studies (415 patients) presented data on toxicity due to chemoradiotherapy in relation to smoking status. Smoking during chemoradiotherapy was associated with increased risk for toxicity (pooled OR: 2.43; 95% CI: 1.43–4.07; Appendix A).

### 3.9. Subgroup Analyses

In subgroup analyses based on radiation treatment techniques (2D/3D conformal radiation therapy (CRT) vs. intensity-modulated radiotherapy (IMRT)/ volumetric modulated arc therapy (VMAT)), we found no statistically significant interaction in terms of LRR (*p*-value = 0.844), DFS (*p*-value = 0.559) or radiation-induced toxicity (*p*-value = 0.325), respectively (Appendix A). As a result, smoking during radiation therapy was associated with shorter LRR and DFS as well as a higher risk for radiation-induced toxicity compared to non-smoking, irrespective of the radiation treatment technique used.

A subgroup analysis of first vs. second/third EGFR TKIs showed that shorter PFS in smokers was evident in both EGFR TKI categories (*p*-value for interaction test = 0.986; Appendix A).

For the pooled analysis of PFS in patients treated with checkpoint inhibitors, we performed a subgroup analysis based on race (Asiatic vs. non-Asiatic population). In both subgroups, smokers were found to have longer PFS compared to non-smokers, and no statistically significant interaction was observed between the two subgroups (*p*-value = 0.801; Appendix A). Similar subgroup analyses based on race were not possible in other treatments or outcomes due to the paucity of data.

In terms of chemotherapy-induced toxicity and the impact of smoking, we were able to perform subgroup analyses for two types of chemotherapy (taxanes and platinum), but we did not find that smoking was associated with the risk of toxicity during either taxane- or platinum-based therapy (Appendix A).

### 3.10. Publication Bias

The risk for publication bias was assessed in three pooled analyses (LRR during radiotherapy; radiation-induced toxicity; PFS during immunotherapy) where an adequate number of studies were included in the pooled analyses. A publication bias in toxicity analyses with potentially missing studies in areas of non-significance cannot be excluded, whereas no evidence on publication bias in the pooled analysis of PFS during EGFR TKIs was observed (Appendix A).

### 3.11. Grading the Evidence According to GRADE

Table 1 presents the certainty of evidence for each pooled analysis according to the GRADE approach. The certainty of evidence varied between very low to moderate. Evidence of smoking and chemotherapy-induced toxicity was the only pooled analysis that was rated as having a very low certainty of evidence.

## 4. Discussion

The present systematic review and meta-analysis summarized the current evidence on the potential impact of smoking during cancer treatment on treatment-related efficacy and toxicity. We found that smoking during radiation therapy or chemoradiotherapy has a negative impact both on treatment-related outcomes such as LRR and DFS and on radiation-induced toxicity with a moderate certainty of evidence. Except for EGFR-TKI therapy, where our findings suggest worse PFS for smokers and for checkpoint inhibitors where smokers seem to derive more benefit compared to non-smokers, smoking appears to have no impact on chemotherapy. However, the certainty of evidence on the impact of smoking on systemic treatment ranged between very low and low, emphasizing the need for further studies on this topic.

Our results confirm the negative impact of smoking on radiation therapy with or without chemotherapy that has been evident in prior meta-analyses dedicated to specific cancer types [14,15]. By use of a broader strategy that included relevant data irrespective of cancer type, our results support the current evidence in a wider perspective. The impact of smoking on radiation therapy is supported by a clear biological rationale through the nicotine-induced re-growth of cancer cells after a single fraction of radiation [23] and the smoking-induced hypoxic microenvironment, which seems to reduce the effect of radiation [24]. The increased risk for radiation-induced toxicity in smokers is in accordance with prior evidence for late toxicity as the main cause of death from second primary lung cancer and cardiovascular events in women previously irradiated for breast cancer [9].

Available evidence to date on the potential impact of smoking on chemotherapy is scarce compared to the evidence on radiation therapy. Although there is a biological rationale for an association between smoking and chemotherapy, mainly through the induction of drug-metabolizing enzymes [17], we were unable to confirm such an association. However, the certainty of evidence for the pooled analyses of smoking and chemotherapy efficacy or toxicity ranged from very low to low, highlighting the need for more research on this question.

Adequate data were available for pooled analyses of two additional types of systemic treatment, namely treatment with EGFR-TKIs in lung cancer and treatment with immunotherapy using PD-1 or PD-L1 inhibitors. Regarding EGFR-TKIs, our results on the negative impact of smoking on treatment efficacy confirm results from prior meta-analyses [25,26], findings which are supported by a biological rationale based on a nicotine-induced resistance mechanism to EGFR-TKIs [27] and a more rapid drug excretion in smokers [28].

Regarding immunotherapy, our pooled analysis suggests that immune checkpoint inhibitors may confer a greater survival benefit on patients who are smokers than those who are non-smokers. Prior meta-analyses based on subgroup analyses of randomized trials imply an improved treatment outcome in smokers treated with immunotherapy compared to non-smokers [29,30]. This paradoxical association, which is confirmed in our pooled analysis including only studies using multivariate analyses to reduce the risk of confounding bias, may be explained by the higher mutational burden (TMB) in tumor cells in smokers [31], which is a potential predictive biomarker for immunotherapy [32] rather than a positive causal effect on outcome directly by smoking. A dose-dependent association between pack years and magnitude of treatment efficacy to checkpoint inhibitors has been recently described [33,34], which is similar to the association between pack years and the level of TMB in tumors [35], further highlighting the potential role of smoking exposure as a surrogate for TMB. Future studies should consider smoking pack years when analyzing the impact of smoking on checkpoint inhibitors.

Our findings should be interpreted in light of the strengths and limitations of the present meta-analysis. In contrast to prior meta-analyses investigating the impact of smoking on specific cancer types or cancer treatment strategies [3,4,5,6,14,15], we aimed to summarize the available evidence on the impact of smoking on cancer treatment regardless of cancer type or treatment strategy. This comprehensive strategy allowed us to address the research question from a broader perspective, thus providing more widely generalizable results. Another strength was the inclusion of studies with results derived from multivariable analyses to limit the risk of confounding by indication or selection bias. This aspect is of particular importance when observational studies are included due to the higher risk of bias in these study types that might lead to over- or underestimation of potential associations [36]. Similarly, we excluded randomized trials that presented results on the association between smoking and treatment efficacy through subgroup analyses to avoid the risk of overestimating the effects [37]. An additional strength was the presentation of the certainty of evidence for each pooled analysis using the GRADE approach, facilitating the implementation of the results in clinical practice.

The meta-analysis has several limitations that need mentioning. For some treatment strategies and outcomes, the number of eligible studies was limited because of the strict criteria used in the selection process. As a result, we were unable to investigate the impact of smoking on outcomes such as pCR or ORR, as well as the influence of smoking on targeted therapies (other than EGFR-TKIs and immunotherapy), brachytherapy, or endocrine therapy. In addition, the vast majority of eligible studies are retrospective, and thus inherent to bias. Another limitation attributable to the limited number of eligible studies was that we were unable to assess the difference in outcomes between former and never smokers since the reference group in our comparisons included both categories. The low number of eligible studies in several pooled analyses limited the possibility of further analyzing the clinical heterogeneity of the results through relevant subgroup analyses. However, we were able to perform some clinically relevant subgroup analyses (e.g., the impact of more modern radiation techniques in outcomes related to radiotherapy) without any evidence for clinical heterogeneity among the tested subgroups. Finally, we could not examine the impact of smoking cessation after treatment on prognosis since the required information was lacking in eligible studies. Future studies should focus on investigating the potential impact of comprehensive smoking cessation programs compared to usual care on the rates of smoking cessation in the oncology setting. Notably, comprehensive smoking cessation programs seem to have similar rates of success in patients with cancer and those without cancer [38].

## 5. Conclusions

In summary, the present meta-analysis confirms a negative impact of smoking during radiation therapy on treatment efficacy and toxicity irrespective of cancer type, with a moderate level of evidence. No similar association between smoking and chemotherapy was observed, but the low certainty of evidence for these comparisons precludes any firm conclusion. Regarding targeted therapies, there is evidence supporting a negative association between smoking and efficacy of EGFR-TKIs and a positive association between smoking and efficacy of checkpoint inhibitors. These data can be used by oncology and radiotherapy staff to give patients more convincing information on the benefits that can be derived from smoking cessation before cancer treatment is commenced. These data could also enable a more balanced discussion on risks and benefits of specific treatment approaches, thus leading to more informed decisions. Besides, the results outline the importance of adopting smoking cessation programs in the oncology setting to support smoking cessation efforts and can serve as a tool for policy makers in designing and promoting smoking cessation strategies. By summarizing current evidence, the present meta-analysis reveals knowledge gaps that can be used to direct further research. In addition to clinical research investigating the impact of smoking on cancer treatment regimens, studies on underlying biological mechanisms are urgently needed to discern causality in observed potential associations.

## Figures and Tables

**Figure 1 cancers-14-04117-f001:**
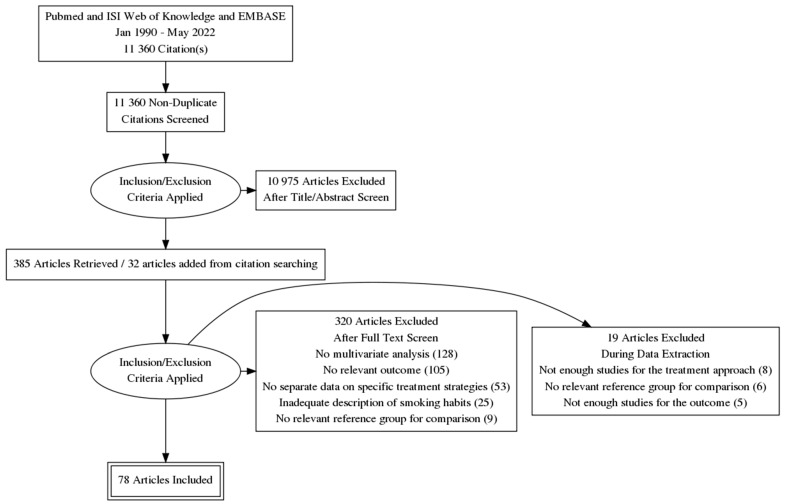
Flowchart diagram of study selection process.

**Figure 2 cancers-14-04117-f002:**
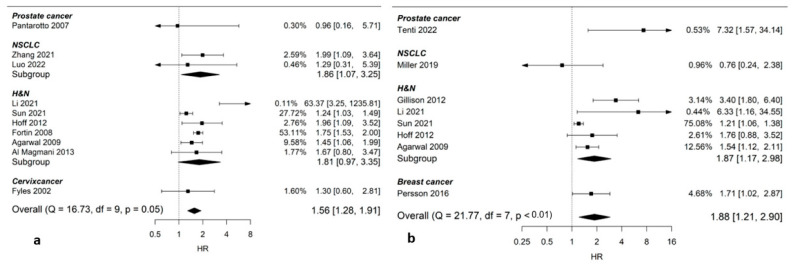
Forest plots on pooled Hazard Ratios (HRs) on smoking during radiotherapy and efficacy. (**a**) Smoking during radiotherapy and locoregional recurrence free survival; (**b**) smoking during radiotherapy and disease-free survival. HR > 1 indicates worse outcome for smokers during radiation treatment, whereas HR < 1 indicates better outcome. Comparison group is non-smoker (former or never). *Abbreviations*: NSCLC, non-small-cell lung cancer; H&N, head and neck cancer.

**Figure 3 cancers-14-04117-f003:**
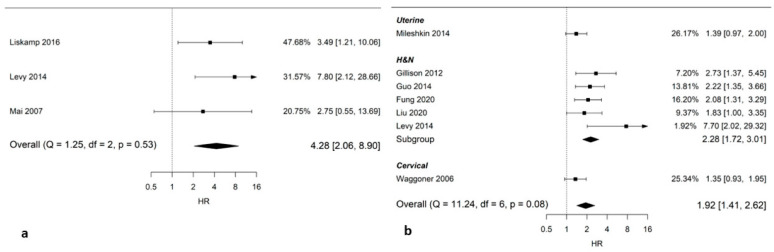
Forest plots on pooled Hazard Ratios (HRs) on smoking during chemoradiotherapy and efficacy. (**a**) Smoking during chemoradiotherapy and locoregional recurrence free survival; (**b**) smoking during chemoradiotherapy and disease-free survival. HR > 1 indicates worse outcome for smokers during radiation treatment, whereas HR < 1 indicates better outcome. Comparison group is non-smoker (former or never). *Abbreviation*: H&N, head and neck cancer.

**Figure 4 cancers-14-04117-f004:**
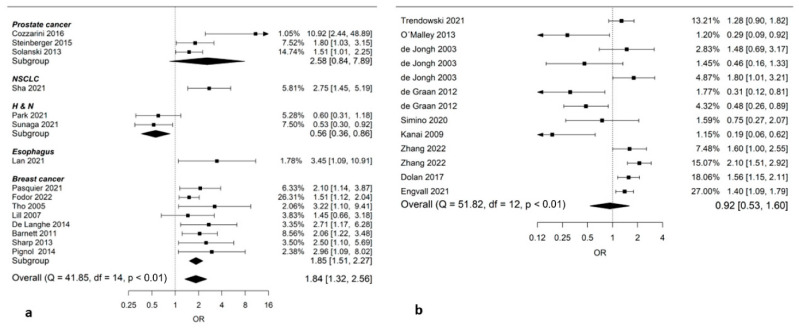
Forest plots on pooled Odds Ratios (ORs) on smoking during treatment and toxicity. (**a**) Smoking during radiotherapy and toxicity; (**b**) smoking during chemotherapy and toxicity. OR > 1 indicates higher risk for toxicity during treatment, whereas HR < 1 indicates lower risk. Comparison group is non-smokers (former or never). *Abbreviations*: NSCLC, non-small-cell lung cancer; H&N, head and neck cancer.

**Table 1 cancers-14-04117-t001:** Certainty of evidence on the impact of smoking on treatment efficacy and toxicity with GRADE approach.

N Studies	Assessment of Evidence						Effect		Certainty of Evidence
	Study Design	Bias	Inconsistency	Indirectness	Imprecision	Other	N Patients	Pooled Effect (95% CI)	
Locoregional recurrence after radiotherapy in patients who smoke versus non-smokers									
10	observation	Serious	Not serious	Not serious	Not serious	No	13276	1.56 (1.28–1.91)	⨁⨁⨁◯ Moderate
Disease-free survival after radiation therapy in patients who smoke versus patients who do not smoke									
9	observation	Serious	Not serious	Not serious	Not serious	No	11185	1.88 (1.21–2.90)	⨁⨁⨁◯ Moderate
Locoregional recurrence after chemoradiotherapy in patients who smoke versus patients who do not smoke									
3	observation	Serious	Not serious	Not serious	Serious	No	339	4.28 (2.06–8.09)	⨁⨁◯◯ Low
Disease-free survival after chemoradiotherapy in patients who smoke versus patients who do not smoke									
7	observation	Serious	Not serious	Not serious	Not serious	No	2096	1.92 (1.41–2.62)	⨁⨁⨁◯ Moderate
Progression-free survival of chemotherapy in patients who smoke vs. non-smoking patients									
6	observation	Serious	Serious	Not serious	Not serious	No	1489	1.22 (0.63–2.36)	⨁⨁◯◯ Low
Progression-free survival of EGFR-TKIs in patients who smoke vs. non-smoking patients									
16	observation	Serious	Not serious	Not serious	Not serious	No	7682	1.46 (1.21–1.77)	⨁⨁⨁◯ Moderate
Progression-free survival of immunotherapy in patients who smoke vs. non-smoking patients									
11	observation	Serious	Not serious	Not serious	Not serious	No	4568	0.70 (0.61–0.82)	⨁⨁⨁◯ Moderate
Radiation-induced toxicity in patients who smoke vs. non-smoking patients									
15	observation	Serious	Not serious	Not serious	Not serious	No	6776	1.84 (1.32–2.56)	⨁⨁⨁◯ Moderate
Chemotherapy-induced toxicity in patients who smoke compared to patients who do not smoke									
9	observation	Serious	Serious	Not serious	Serious	No	3307	0.92 (0.53–1.60)	⨁◯◯◯ Very low
Chemoradiotherapy-induced toxicity in patients who smoke vs. non-smoking patients									
4	observation	Serious	Not serious	Not serious	Serious	No	415	2.43 (1.45–4.07)	⨁⨁◯◯ Low

*Abbreviations*: N, number; CI, Confidence Interval.

## Data Availability

The data presented in this study are available on request from the corresponding author.

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
