# Peer review of "Effect of Smoking on Treatment Efficacy and Toxicity in Patients with Cancer: A Systematic Review and Meta-Analysis"

_cancers, 2022, doi:10.3390/cancers14174117_

Round 1

Reviewer 1 Report

Thank you very much for your re-submission Points in bold are my major concerns.

1. It is nice to see authors have included more updated articles from more different sources with a registration of PROSPERO. Last time the authors have included a PRISMA checklist, however, I couldn’t see this in this updated version.

2. Authors stated in the abstract and line 176 that there are 97 eligible articles. However, in Figure 1, “384 articles retrieved / 32 articles added from citation search” (i.e. 416 in total); Also, “320 articles excluded”. 416-320=96, not 97. Please kindly cross-check the figures.

3. I understand that there may be a difficulty in performing sensitivity analyses due to limited number of studies last time. This time the authors have tried to conduct some e.g. re-analysis with the exclusion of RCT. In addition to cancer types, subgroup analyses by RT technique, generation of were also performed to make the article more comprehensive. For the subgroup analyses, I would like to see the forest plots showing the odds ratio in the supplementary materials.

4. Please include a column of follow-up time of each study in supplementary Table S1 as suggested last time. Too short follow-up is particularly problematic for radiation therapies. It is an important information for readers to judge whether the studies showed adequate estimation of the treatment efficacy.

5. Line 339. I strongly suggest not to say “smoking was suggested to improve the treatment efficacy of checkpoint inhibitors” as it may have an implication of encouraging smoking. Same problem in line 310-311. Authors are suggested to say “immune checkpoint inhibitors may have a greater survival benefit in patients who are smokers than those in non-smokers.”

6. Please further elaborate the implications of your study (apart from only mentioning further studies in line 382-386). What are the other potential applications of your study results? Clinical decision-making and healthcare advice to patients? Policy-making? Design of health promotion programmes?

7. Please make sure all abbreviations are provided in ALL of the figures and tables in the main text AND in the supplementary materials.

Round 2

Reviewer 1 Report

All comments are well addressed.

This manuscript is a resubmission of an earlier submission. The following is a list of the peer review reports and author responses from that submission.

Round 1

Reviewer 1 Report

A well written paper that does summarize and analyze the available literature on the impact of continued smoking while receiving the different classic treatments for cancer.

I would recommend that the author discuss the implications of their work, in particular the main deficiency we face is the reliance on retrospective data and the almost lack of prospective randomized trials, as they noted in their methods section: "Of 70 eligible studies, 56 (80%) were retrospective, 12 (17%) were prospective and 2 (3%) were randomized studies studies.

It has been noted by others that one of the main obstacles to overcome this issue in our field is in part due to the ethical difficulty about delaying someone's treatment until they quit smoking. However, what can be done ethically and in a scientifically sound way is to randomize patients to the usual approach in terms of advice to quit and the other group to a comprehensive approach to quit. Which would show the improvement or lack of form quitting. The other obstacle towards progress in this area is the notion that treating tobacco treatment in cancer setting is not effective. Important to note: Our group have shown and published the evidence from that a comprehensive approach to treat tobacco does lead to significantly higher abstinence from smoking (Cinciripini, et al. 2019) 

Addressing these two points above would give the paper significance and impact in hopes to move the field forward.

Author Response

Reviewer 1

A well written paper that does summarize and analyze the available literature on the impact of continued smoking while receiving the different classic treatments for cancer.

I would recommend that the author discuss the implications of their work, in particular the main deficiency we face is the reliance on retrospective data and the almost lack of prospective randomized trials, as they noted in their methods section: "Of 70 eligible studies, 56 (80%) were retrospective, 12 (17%) were prospective and 2 (3%) were randomized studies studies.

It has been noted by others that one of the main obstacles to overcome this issue in our field is in part due to the ethical difficulty about delaying someone's treatment until they quit smoking. However, what can be done ethically and in a scientifically sound way is to randomize patients to the usual approach in terms of advice to quit and the other group to a comprehensive approach to quit. Which would show the improvement or lack of form quitting. The other obstacle towards progress in this area is the notion that treating tobacco treatment in cancer setting is not effective. Important to note: Our group have shown and published the evidence from that a comprehensive approach to treat tobacco does lead to significantly higher abstinence from smoking (Cinciripini, et al. 2019)

Addressing these two points above would give the paper significance and impact in hopes to move the field forward.

Response to Reviewer 1:

We would like to thank the reviewer for the comments. We have added two sentences in the Discussion section to discuss the issues raised by the reviewer.

Reviewer 2 Report

Thank you very much for your interesting work. Authors tried to explore the effect of smoking on treatment efficacy and toxicity in patients with cancer. However, I have the following major concerns for authors to address before the publication is re-considered.

  1. The source of data of this meta-analysis was mainly limited to two databases which I think is far from enough. There are some actually other databases e.g. EMBASE and authors may miss some other important studies.
  2. Authors did not mention the number of patients included and median follow-up time in the abstract.
  3. Line 72. There is a typo “searchusing”.
  4. Search strategy is too simple. Suggest to provide a detailed step-by-step searching strategy in different databases and ALL of the keywords in the supplementary materials.
  5. It is understood that studies with only univariable Cox hazards model were not included in this meta-analysis. However, I think these studies may also provide prognostic information and I suggest to include these at least in the systematic review.
  6. Line 23 Data extraction. Authors should state what outcomes to be retrieved i.e. -- the hazard ratio (HR) and 95% CI associated with the respective smoking status and outcome.
  7. More elaboration is required on the part of “Grading the certainty of evidence” e.g. the criteria on the level of evidence certainty.
  8. It is a common practice for a robust meta-analysis to include sensitivity analyses. However, authors did not conduct any test to ensure the consistence of study results. Examples could be stratifying retrospective studies and RCTs.
  9. Subgroup analysis is also important to make the whole article more comprehensive. Apart from the analysis stratified by cancer types, authors could conduct analysis stratified by e.g. regions, ethnicity, gender or age.
  10. Publication bias was not assessed in this study.
  11. It seems that authors did not perform heterogeneity tests to justify the use of the random-effects model. Usually random-effects model is used when there is significant study heterogeneity demonstrated by I-square and P value, otherwise, fixed-effects model should be adopted. Personally I think there should be sort of heterogeneity due to the difference in ethnicity, regions, use of different radiation technique, radiation dose, etc. Authors should address these issues by certain statistical tests.
  12. Line 130. Treatment-related toxicities of grade 3 or above is considered severe and is usually included in the meta-analysis for the toxicity outcomes. Authors stated “grade 2” which I think is clinically and academically not suitable.
  13. Line 131. Which version of CTCAE did the authors refer to? How did authors adjust for the difference of toxicity defined by different versions of CTCAE in different studies which were conducted in different periods?
  14. The mathematic formula in “data synthesis” help readers understand more but I suggest to move these to the supplementary materials.
  15. Authors did not consider the important endpoint of “cancer-specific survival” which reflects the long term efficacy of treatment.
  16. There are only a few studies included in EACH analysis for different endpoints. However, authors stated that this meta-analysis consisted of 70 studies and this is misleading.
  17. Supplementary Table 1. Authors should include follow-up time, treatment modality, number of smokers, regions, outcomes included and their reported or calculated HR and OR and confidence intervals of each study.
  18. A toxicity profile including the incidence of some common and specific adverse events is highly appreciated.
  19. Supplementary Table 1. Authors stated at the beginning and in the Results that they have 70 studies in this meta-analysis, but from the column of “Inclusion in meta-analysis”, some studies were actually not included. Please provide a brief reason of exclusion next to this column. The total number of studies considered in this meta-analysis should NOT be 70 if these are really excluded (there are 18 studies stated with “no”).
  20. Kindly provide the abbreviations in ALL supplementary figure and figures.
  21. Authors did not address the effect of ex-smoking (though ex-smokers are considered as non-smoker in this analysis) which is another very important prognostic factor.
  22. Cumulative pack-years was not considered. Thought there may be limited number of studies, related information should be presented and discussed qualitatively.
  23. Did the authors register their systematic review in PROSPERO?
  24. Authors mentioned that one of their strengths is that they are able to “summarize the available evidence on the impact of smoking on cancer treatment regardless of cancer type or treatment strategy”. However, due to limited studies included in each of the analysis leading to a relatively underpowered results, this is indeed a study limitation, rather than a strength. On the other hand, the radiation technique has gradually changed from 2D/3DCRT to IMRT in the past decade resulting in better tumour response in general. It seems that authors did not consider this important issue.
  25. Discussion “we excluded randomized trials that presented results on the association between smoking and treatment efficacy through subgroup analyses to avoid the risk of overestimating the effects”. I suggest authors to include these studies into their meta-analysis and conduct subsequent sensitivity analyses after the removal of these.
  26. Authors did not mention the implications of this study e.g. how does their study help clinical decision and even policy making? Any articulated programmes could be recommended like smoking cessation?
  27. Conclusion is better to be shortened.
  28. Figure 1 Flowchart diagram. The part “reports excluded” are not in English.
  29. Please also provide the definitions of all endpoints.

Author Response

Reviewer 2

Thank you very much for your interesting work. Authors tried to explore the effect of smoking on treatment efficacy and toxicity in patients with cancer. However, I have the following major concerns for authors to address before the publication is re-considered.

We would like to thank the reviewer for the valuable comments. Our point-by-point response can be seen in bold.

  1. The source of data of this meta-analysis was mainly limited to two databases which I think is far from enough. There are some actually other databases e.g. EMBASE and authors may miss some other important studies.

We thank the reviewer for this comment. We have included EMBASE as well in the searching strategy without finding any additional study. The flowchart diagram has been updated.

  1. Authors did not mention the number of patients included and median follow-up time in the abstract.

The number of patients included depends on the number of eligible studies in each pooled analysis and this is the reason why we chose to mention this information in the main text for each pooled analysis. There is no uniform median follow-up time for the whole study either considering the different outcomes that have been analyzed.

  1. Line 72. There is a typo “searchusing”.

The typo has been corrected.

  1. Search strategy is too simple. Suggest to provide a detailed step-by-step searching strategy in different databases and ALL of the keywords in the supplementary materials.

We thank the reviewer for this comment. The searching strategy described in the main text is only an overview of the strategy and not all the steps.

  1. It is understood that studies with only univariable Cox hazards model were not included in this meta-analysis. However, I think these studies may also provide prognostic information and I suggest to include these at least in the systematic review.

We thank the reviewer for this comment. However, we think that the inclusion of results from univariate analyses would introduce a higher risk of bias to our pooled results due to their uncertainty.

  1. Line 23 Data extraction. Authors should state what outcomes to be retrieved i.e. -- the hazard ratio (HR) and 95% CI associated with the respective smoking status and outcome.

We thank the reviewer for this comment. We have added this information.

  1. More elaboration is required on the part of “Grading the certainty of evidence” e.g. the criteria on the level of evidence certainty.

We thank the reviewer for this comment. We have chosen to use a reference to the GRADE framework for the readers who would like to read more about GRADE. The GRADE approach is a well-established approach for grading the certainty of evidence and we do not feel that there is any reason to explain in more details.  

  1. It is a common practice for a robust meta-analysis to include sensitivity analyses. However, authors did not conduct any test to ensure the consistence of study results. Examples could be stratifying retrospective studies and RCTs.

We thank the reviewer about this comment. Indeed, sensitivity analyses are often a part of a meta-analysis. However, the choice to perform a sensitivity analysis or not is based on the number of eligible studies and the validity of a sensitivity analysis. Unfortunately, the number of eligible studies in the pooled analyses precludes any meaningful sensitivity analysis.

  1. Subgroup analysis is also important to make the whole article more comprehensive. Apart from the analysis stratified by cancer types, authors could conduct analysis stratified by e.g. regions, ethnicity, gender or age.

The present meta-analysis faces the same problem as stated above that precludes the performance of a sensitivity analysis or a subgroup analysis. As the reviewer mentioned, we have already performed a subgroup analysis of relevance regarding the cancer type (in some of the pooled analyses) but we are lacking adequate data for other subgroup analyses of potential interest.

  1. Publication bias was not assessed in this study.

We thank the reviewer for this comment. We have added contour-enhanced funnel plots (as supplementary material) to screen for publication bias in three of our meta-analytic models with adequate number of studies. For models with low number of studies, funnel plots or tests for publication bias are not recommended (Sterne et al 2011). Panels A and C suggest possible evidence of publication, though even here, the sample size is on the boundary of the recommended number of studies to assess for publication bias. Further studies would be required to make a more rigorous assessment of publication bias.

  1. It seems that authors did not perform heterogeneity tests to justify the use of the random-effects model. Usually random-effects model is used when there is significant study heterogeneity demonstrated by I-square and P value, otherwise, fixed-effects model should be adopted. Personally I think there should be sort of heterogeneity due to the difference in ethnicity, regions, use of different radiation technique, radiation dose, etc. Authors should address these issues by certain statistical tests.

We thank the reviewer for bringing up this important aspect. The choice between fixed and random effects models should not be based on assessment of heterogeneity. Indeed, this is a common misconception (both the fixed and random effects models can be used in the presence of heterogeneity). Rather we pre-specified random effects models due to the nature of our statistical inference; that being unconditional inference which can be generalized to some hypothetical population of studies. This is supportive of our choice of random effects models since we assume that the true effect sizes will vary between studies, due to the variables indicated above such as ethnicity, region, radiation technique and dose etc. An explanation of this reasoning can be found in the accepted answer here (from the author of metaphor R package which we used in our study):

https://stats.stackexchange.com/questions/156603/how-to-choose-between-fixed-effects-and-random-effects-model-in-meta-analysis

Q tests for heterogeneity were already displayed on our forest plots.

  1. Line 130. Treatment-related toxicities of grade 3 or above is considered severe and is usually included in the meta-analysis for the toxicity outcomes. Authors stated “grade 2” which I think is clinically and academically not suitable.

We thank the review for this comment. We are not sure that we understand exactly what the reviewer would expect us to do. In general, a grade 2 toxicity refers to a toxicity influencing patients’ ADL and needs to be treated from physicians. So we think that including grade 2 toxicity to our analyses is both clinically and academically suitable.

  1. Line 131. Which version of CTCAE did the authors refer to? How did authors adjust for the difference of toxicity defined by different versions of CTCAE in different studies which were conducted in different periods?

We performed no adjustment for different CTCAE versions but we accepted any version of CTCAE. We have added this information in the Methods.

  1. The mathematic formula in “data synthesis” help readers understand more but I suggest to move these to the supplementary materials.

We thank the reviewer for this comment. We moved the formulas to the supplementary materials as suggested.

  1. Authors did not consider the important endpoint of “cancer-specific survival” which reflects the long term efficacy of treatment.

We thank the reviewer for this comment. We chose not to consider cancer specific survival in our meta-analysis since this outcome can be influenced by the treatment approaches in general rather than a specific treatment, which was the scope of our study. For instance, the risk of cancer specific survival in a patient with NSCLC treated with EGFR-directed therapy would reflect the impact of subsequent therapies as well and not only the impact of EGFR-directed therapy.

  1. There are only a few studies included in EACH analysis for different endpoints. However, authors stated that this meta-analysis consisted of 70 studies and this is misleading.

We thank the reviewer for this comment. We have changed this information throughout the text.

  1. Supplementary Table 1. Authors should include follow-up time, treatment modality, number of smokers, regions, outcomes included and their reported or calculated HR and OR and confidence intervals of each study.

We thank the reviewer for this comment. We have added to Supplementary Table 1 some information of interest for the readers (treatment modality, number of smokers, regions). The table includes already information on outcome. However, we cannot see how follow-up time would be of relevance considering that many studies report outcomes where time is not important as toxicity so we decided not to include this information.

  1. A toxicity profile including the incidence of some common and specific adverse events is highly appreciated.

We thank the reviewer for this comment. However, we think that the addition of this information would be beyond the scope of our review.

  1. Supplementary Table 1. Authors stated at the beginning and in the Results that they have 70 studies in this meta-analysis, but from the column of “Inclusion in meta-analysis”, some studies were actually not included. Please provide a brief reason of exclusion next to this column. The total number of studies considered in this meta-analysis should NOT be 70 if these are really excluded (there are 18 studies stated with “no”).

We thank the reviewer for this comment. We have included this information in the new Flowchart diagram.

  1. Kindly provide the abbreviations in ALL supplementary figure and figures.

We thank the reviewer for this comment. We have added the abbreviations to all figures and tables in supplementary material.

  1. Authors did not address the effect of ex-smoking (though ex-smokers are considered as non-smoker in this analysis) which is another very important prognostic factor.

We thank the reviewer for raising this important aspect. Indeed, we were unable to perform pooled analyses for ex-smokers compared to non-smokers due to the limited number of studies. This aspect has been mentioned as a limitation in the limitation section of the Discussion.  

  1. Cumulative pack-years was not considered. Thought there may be limited number of studies, related information should be presented and discussed qualitatively.

This information was not included in the vast majority of the eligible studies and this is the reason why we did not mention this aspect in our review.

  1. Did the authors register their systematic review in PROSPERO?

This information has been added to the text.

  1. Authors mentioned that one of their strengths is that they are able to “summarize the available evidence on the impact of smoking on cancer treatment regardless of cancer type or treatment strategy”. However, due to limited studies included in each of the analysis leading to a relatively underpowered results, this is indeed a study limitation, rather than a strength. On the other hand, the radiation technique has gradually changed from 2D/3DCRT to IMRT in the past decade resulting in better tumour response in general. It seems that authors did not consider this important issue.

We thank the reviewer for this comment. We totally agree that the limited studies in specific pooled analyses is a major drawback of our study and this has been acknowledged to the limitations. We have changed the text in the discussion section regarding the radiation techniques.

  1. Discussion “we excluded randomized trials that presented results on the association between smoking and treatment efficacy through subgroup analyses to avoid the risk of overestimating the effects”. I suggest authors to include these studies into their meta-analysis and conduct subsequent sensitivity analyses after the removal of these.

We thank the reviewer for this comment. However, we decided to keep these results outside of our pooled analysis considering the high risk for bias in such analyses that would result in misleading results even in a subgroup analysis.

  1. Authors did not mention the implications of this study e.g. how does their study help clinical decision and even policy making? Any articulated programmes could be recommended like smoking cessation?

We thank the reviewer for this comment. The implication of our work has been described as a part of the conclusion. We have added a sentence about the implication of this study in policy makers.

  1. Conclusion is better to be shortened.

We thank the reviewer for this comment. The Conclusion session includes the implication of our work and this is the reason why this part is somewhat extensive. We prefer to keep this part as it is since we believe that the implication of our work is more suitable to be presented to this section.

  1. Figure 1 Flowchart diagram. The part “reports excluded” are not in English.

We thank the reviewer for this comment and apologize for this. We have changed the Flowchart diagram.

  1. Please also provide the definitions of all endpoints.

We have added a sentence explaining our approach about definition of endpoints.

Round 2

Reviewer 2 Report

Although the flow of the manuscript after revisions has been improved, due to underpowered results, lack of sensitivity tests and heterogeneity tests, I am afraid  that the conclusion of this study may not be the valid one.

Author Response

Although the flow of the manuscript after revisions has been improved, due to underpowered results, lack of sensitivity tests and heterogeneity tests, I am afraid that the conclusion of this study may not be the valid one.

We thank the reviewer for the comment. We have performed a new searching of the databases (the last searching on the prior version was on February 2021) to include the latest studies on this topic. The addition of studies from the past year increased the number of eligible studies and consequently the power of our statistical analyses. It enabled us also to perform some new analyses of interest.

The lack of sensitivity analyses and subgroup analyses were due to the lack of enough studies to perform such analyses rather than our refusal to perform them. After inclusion of one more year in our searching strategy, we were able to proceed to some additional analyses of clinical interest.

The heterogeneity test is present in ALL the pooled analyses (it was present already from the 1st version) but it is incorporated in the forest plot. In the bottom left side of each pooled analysis, you can find the heterogeneity test expressed as Q and p-value. We have added this information in the Methods as well.